# Contracting CAG/CTG repeats using the CRISPR-Cas9 nickase

Cinzia Cinesi[1], Lorène Aeschbach[1], Bin Yang[1] & Vincent Dion[1]

CAG/CTG repeat expansions cause over 13 neurological diseases that remain without a cure. Because longer tracts cause more severe phenotypes, contracting them may provide a therapeutic avenue. No currently known agent can specifically generate contractions. Using a GFP-based chromosomal reporter that monitors expansions and contractions in the same cell population, here we find that inducing double-strand breaks within the repeat tract causes instability in both directions. In contrast, the CRISPR-Cas9 D10A nickase induces mainly contractions independently of single-strand break repair. Nickase-induced contractions depend on the DNA damage response kinase ATM, whereas ATR inhibition increases both expansions and contractions in a MSH2- and XPA-dependent manner. We propose that DNA gaps lead to contractions and that the type of DNA damage present within the repeat tract dictates the levels and the direction of CAG repeat instability. Our study paves the way towards deliberate induction of CAG/CTG repeat contractions *in vivo*.

[1] Center for Integrative Genomics, University of Lausanne, 1015 Lausanne, Switzerland. Correspondence and requests for materials should be addressed to V.D. (email: Vincent.dion@unil.ch).

Repetitive DNA sequences are hotspots for genome instability because they pose a particular challenge to the DNA repair and replication machineries. Their mutation often leads to disease[1]. For example, tracts of CAG/CTG triplets (henceforth referred to as CAG repeats) longer than about 35 units cause at least 14 different currently incurable neurological and neuromuscular diseases[2]. In addition, when CAG repeats expand to pathological lengths, they become highly dynamic and their length changes at high frequencies in both somatic and germ cells throughout the lifetime of an individual[3–6].

The molecular mechanisms governing CAG repeat instability revolve around the ability of these sequences to fold into non-B-DNA structures when exposed as single-stranded DNA[7–9]. These unusual structures are mistaken for damaged DNA whether or not they contain lesions. The subsequent repair is error-prone due to the repetitive nature of the sequences and their structure-forming ability[3]. Another non-mutually exclusive model suggests that DNA damage within the repeat tract triggers repair, which is, in turn, error-prone due to secondary structures formed by these sequences[5]. In support of these models, several DNA repair pathways promote the instability of expanded CAG repeats, including mismatch repair[10], double-strand break (DSB) repair[11–14], transcription-coupled nucleotide excision repair[15,16], base excision repair (BER)[17,18], as well as DNA replication[19]. In contrast, single-strand break (SSB) repair (SSBR)[20] and signalling via the DNA damage response (DDR)[21] antagonize CAG repeat instability. Therefore, changes in repeat length provide an opportunity to understand the interaction and interdependence of several different DNA repair pathways at naturally occurring sequences.

Importantly, repeat length determines in large part the severity of the diseases caused by expanded repeats[4]. It has therefore been proposed that contracting the repeat tract would be beneficial in reducing phenotype expression. Repeat expansion, on the other hand, would further exacerbate the disease symptoms[4,22,23]. Currently, there is no treatment that specifically shrinks CAG repeats. This is, in part, because the assays used to measure repeat instability are tedious, slow and/or can only survey instability in one direction. Consequently, the understanding of the mechanism of CAG repeat instability remains poor. Elucidating how contractions can be induced, without also provoking expansions, is critical in designing therapeutic avenues.

Here we present a green fluorescent protein (GFP)-based chromosomal reporter assay that can monitor both CAG repeat expansions and contractions in the same human cell population. We combined this assay with gene-editing tools, namely zinc finger nucleases (ZFNs) and the CRISPR-Cas9 technology. We found that DSBs induced within the expanded repeat tract either by the Cas9 nuclease or a ZFN led to both expansions and contractions. Remarkably, the Cas9 D10A mutant (referred to as the Cas9 nickase) induces instability with a marked bias towards contractions and no detectable off-target mutations. We implicate the DDR kinases ataxia telangiectasia mutated (ATM) and ataxia telangiectasia and Rad3 related (ATR) in promoting contractions and preventing instability, respectively. Moreover, it is not dependent on the SSBR factors XRCC1 and PARylation. Cas9 nickase-induced repeat contraction appears to occur via a pathway different from SSBR- or BER-induced instability. We propose that DNA gaps may be the crucial mutagenic intermediate during nickase-induced contractions. Our results have important implications for gene editing in expanded trinucleotide repeat diseases.

## Results

**A GFP-based assay to detect CAG repeat instability.** We made use of a recently described GFP-based assay capable of detecting contractions in human cells[24] (Fig. 1a). In this assay, CAG repeats within the intron of a GFP mini-gene interfere with splicing in a repeat length-dependent manner, with longer repeats diminishing GFP production. Thus, GFP intensities, measured by flow cytometry, serve as a proxy for the length of the repeat tract (Supplementary Fig. 1A,B). The reporter is present as a single copy integrated in the genome of human HEK293 T-Rex Flp-In cells. Its transcription is driven by a doxycycline (dox)-inducible promoter. A second isogenic cell line, GFP(CAG)$_0$, harbours the same reporter at the same genomic location but is devoid of a CAG repeat. Santillan et al.[24] validated the assay by expressing a ZFN that cuts the CAG repeat tract. This treatment increased the number of cells with higher GFP intensities (GFP$^+$) in a reporter cell line with 89 repeats (GFP(CAG)$_{89}$) by about 3.5-folds, suggesting that the ZFN treatment induced contractions. They did not report testing for expansions.

To determine whether we could monitor expansions using this assay, we sorted GFP$^-$ and GFP$^+$ cells from a population of GFP(CAG)$_{101}$ cells using fluorescence-activated cell sorting (FACS). We defined GFP$^-$ cells as those within the 1% of the cells in the population expressing the least amount of GFP. Similarly, GFP$^+$ cells are the brightest 1% in the population. From the GFP$^-$ population, we isolated 19 clones with expansions reaching up to 258 CAGs (Supplementary Fig. 1C). Of the 12 GFP$^+$ clones isolated, 11 had contractions, the largest of which shrank the repeat tract down to 33 CAGs. The allele sizes in GFP$^-$ and GFP$^+$ cells were significantly different ($P = 1.0 \times 10^{-5}$, using a Wilcoxon $U$-test), demonstrating that repeat size differences can be detected with this assay. Sequencing the region flanking the CAG repeats also uncovered deletions in five single clones with contractions (Supplementary Fig. 1D). With the exception of one clone that contained a complex rearrangement, the clones with deletions included 2 bp of microhomology at the junction, suggesting that a minor CAG repeat instability pathway is due to an error-prone alternative end-joining mechanism, as suggested recently[25]. Similar results were obtained after sorting cells from populations that were kept in culture for 6 months with or without dox (Supplementary Fig. 1E–H). These results demonstrate that the assay can detect contractions as well as expansions that nearly triple the size of the repeat tract.

**DSBs induce both contractions and expansions.** To determine whether ZFN-induced expansions in addition to the contractions reported by Santillan et al.[24], we first repeated the same experiment. Here we defined GFP$^-$ and GFP$^+$ cells as those with GFP intensities in the brightest and dimmest 1% after transfection with the control vector, (Supplementary Fig. 2A—see Methods). We reproduced their results: ZFN expression increased the frequency of GFP$^+$ cells by 3.2-folds, but had no effect on the number of GFP$^-$ cells (Fig. 1b). While optimizing the assay, we noted that GFP intensities increased on the addition of dox for 72 h before reaching a steady-state level (Supplementary Fig. 2B). This is in contrast to the 24 h previously reported[24]. Increasing the time of GFP induction raised the overall apparent average intensity of GFP and unmasked an additional GFP$^-$ cell population only in the sample transfected with both ZFN arms (Fig. 1c). This approach revealed 2.5- and 3.9-fold increases in the proportion of GFP$^-$ and GFP$^+$ cells, respectively, on expression of both ZFN arms compared with transfecting an empty vector control (Fig. 1d). Expressing either ZFN arm individually led to only small changes in GFP levels: between 1.3- and 1.4-fold increases in the number of GFP$^-$ cells and between 0.9- and 1.5-fold for GFP$^+$ cells (Fig. 1d). Expressing both ZFN arms in the GFP(CAG)$_0$ cell line had no effect on GFP intensities (Supplementary Fig. 2C), confirming

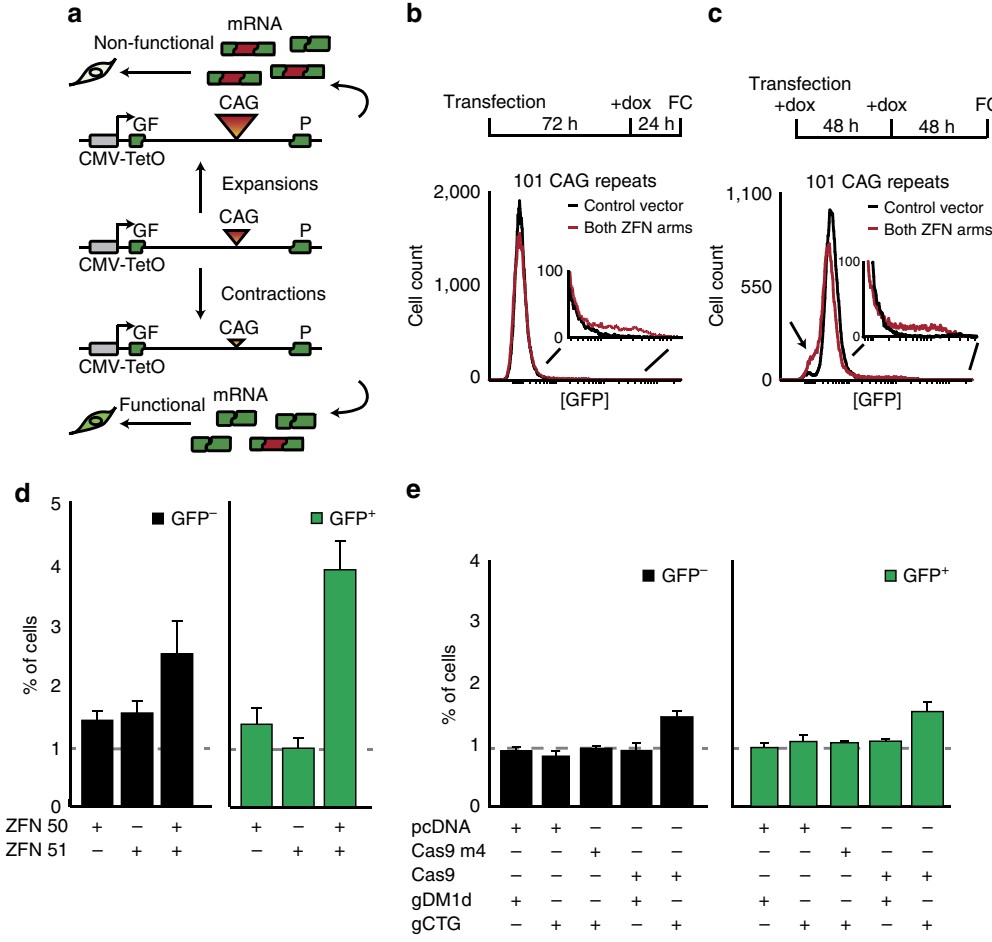

**Figure 1 | DSBs within CAG repeats lead to expansions and contractions.** (**a**) GFP-based assay to detect changes in repeat length. (**b**) Representative flow cytometry profiles after expression of a ZFN in GFP(CAG)$_{101}$ using the protocol from ref. 24. (**c**) Representative flow cytometry profiles with increased dox induction time uncovering an increase in GFP$^-$ cells on ZFN expression in GFP(CAG)$_{101}$ cells (arrow). (**d**) Quantification of the ZFN experiments in **c** revealed that ZFN induces the appearance of GFP$^-$ and GFP$^+$ cells. ZFNs are composed of two different ZFN arms, each fused to a FokI nuclease that must dimerize to be active. ZFN 50 and ZFN 51 are individual ZFN arms[24]. The dashed line represents the number of cells present in gates set to include the dimmest (GFP$^-$) or brightest (GFP$^+$) 1% of the cells when a control vector, pcDNA3.1 Zeo, is transfected. Error bars are s.e.m. from 15 replicates for experiments with both ZFN arms, 12 for the single ZFN transfections. (**e**) Quantification of GFP$^-$ and GFP$^+$ cells obtained after expression of the indicated vectors. Dashed line: dimmest (GFP$^-$) or brightest (GFP$^+$) 1% of the cells transfected with the Cas9 nuclease vector and the empty gRNA plasmid, pPN10. The error bars are s.e.m. Number of replicates per treatment: pcDNA + gDM1d, $n = 3$; pcDNA + gCTG, $n = 5$; Cas9 m4 + gCTG, $n = 4$; Cas9 + gDM1d, $n = 3$; Cas9 + gCTG, $n = 7$. FC, flow cytometry; dox, doxycycline.

that the presence of the repeat tract is necessary. We confirmed that GFP$^-$ cells contained expansions and GFP$^+$ cells harboured contractions by sorting cells exposed to both ZFN arms. Of the 9 GFP$^-$ clones analysed, 8 revealed an expansion (Supplementary Fig. 2D,E). None of them contained deletions and were therefore not GFP$^-$ because they had lost the GFP reporter. Of the 13 GFP$^+$ clones, 11 had contractions. Of those, 3 had deletions in the flanking sequences, which is similar to the findings of a previous study constrained to measuring only contractions and using a different ZFN[26]. Here again, the size of GFP$^-$ and GFP$^+$ cells in the recovered clones were significantly different ($P = 5 \times 10^{-4}$, using a Wilcoxon $U$-test). These results demonstrate that GFP$^-$ and GFP$^+$ cells accurately reflect the presence of expansions and contractions, making this assay especially well suited to detect expansions and contractions quickly within a chromosomal environment.

To confirm that DSBs within the repeat tract lead to both expansions and contractions, we used a second type of programmable nuclease: CRISPR-Cas9. This bacterial nuclease

is guided to virtually any sequence of interest by a guide RNA (gRNA) molecule, where it induces blunt-ended DSBs, making it a highly effective gene-editing tool[27–29]. Transfection of a vector expressing a gRNA that targets the unrelated *DMPK* locus (gDM1d) together with Cas9 did not affect GFP expression (Fig. 1e). Similarly, expressing a gRNA containing six CTGs as the target sequence (gCTG) alone, expressing the Cas9 nuclease plus the gCTG in GFP(CAG)$_0$ cells, or the gCTG together with a catalytically inactive version of Cas9 (Cas9m4) did not change GFP expression significantly (Fig. 1e and Supplementary Fig. 2F). However, expressing the Cas9 nuclease together with gCTG resulted in a meek 1.4- and 1.5-fold induction of GFP$^-$ and GFP$^+$ cells, respectively, compared with co-transfecting the Cas9 expression vector with the empty gRNA vector (Fig. 1e). This low efficiency may reflect that the protospacer adjacent motif next to the target sequence of gCTG is not the canonical NGG. We conclude that DSBs induced within the repeat tract by a ZFN or the Cas9 nuclease provoke nearly as many expansions as contractions.

**The Cas9 nickase induces mainly CAG repeat contractions.** The use of the Cas9 enzyme allowed us to test whether the type of DNA damage present within the repeat tract influences CAG repeat instability. The Cas9 D10A mutant can be used with the same gRNA to introduce DNA nicks on the strand complementary to the gRNA[30]. DNA nicks are important intermediates in repeat instability *in vitro*[31,32]. We therefore asked whether inducing DNA nicks with the Cas9 nickase could influence CAG repeat instability.

We found that expressing the Cas9 nickase together with gCTG in GFP(CAG)$_{101}$ cells increased the number of GFP$^-$ cells by 1.6-fold and GFP$^+$ cells by 3.2-folds compared with cells expressing only the nickase (Fig. 2a). Transfecting the Cas9 nickase with gCAG, which cuts the opposite strand compared with gCTG, had a similar effect, leading to increases of 1.4- and 3.7-folds in GFP$^-$ and GFP$^+$ cells, respectively (Fig. 2a). To control for potential indirect effects on GFP expression, we expressed the Cas9 nickase along with gDM1d. This had no effect on GFP expression (Fig. 2a). In addition, the gCTG alone did not increase either GFP$^-$ or GFP$^+$ cells, similar to expressing the gCTG together with the Cas9m4 mutant (Fig. 1e), suggesting that the activity of the nickase is necessary. Increasing the number of

transfections to three in the span of 12 days further increased the number of GFP$^+$ cells to 6.2-folds, without a concomitant change in GFP$^-$ cells (Fig. 2b and Supplementary Fig. 2G,H; $P = 0.32$ and 0.001 for GFP$^-$ and GFP$^+$ cells, respectively, using a Wilcoxon $U$-test). The Cas9 nickase did not increase the number of dead cells, which could skew the quantification of GFP$^-$ and GFP$^+$ cells (Supplementary Table 1). Also, the difference in the number of GFP$^+$ cells induced between the nuclease and the nickase was not due to differences in expression levels of the Cas9 enzyme (Supplementary Fig. 3A,B). We further confirmed that the way we quantified the data did not induce a bias against expansions (Supplementary Fig. 3C,D). These observations suggest that the Cas9 nickase leads to instability with a bias towards contractions.

To confirm this effect using an assay that is independent of the GFP reporter, we isolated DNA after expressing the Cas9 nickase and gCTG together and performed small-pool PCR (SP-PCR). This method bypasses the inherant advantage in amplification efficiency that smaller alleles have by setting up a larger number of reactions, each with only a few genomes as templates[33]. Using samples treated according to our 12-day regimen, we could detect larger and more frequent contractions in cells exposed to both the

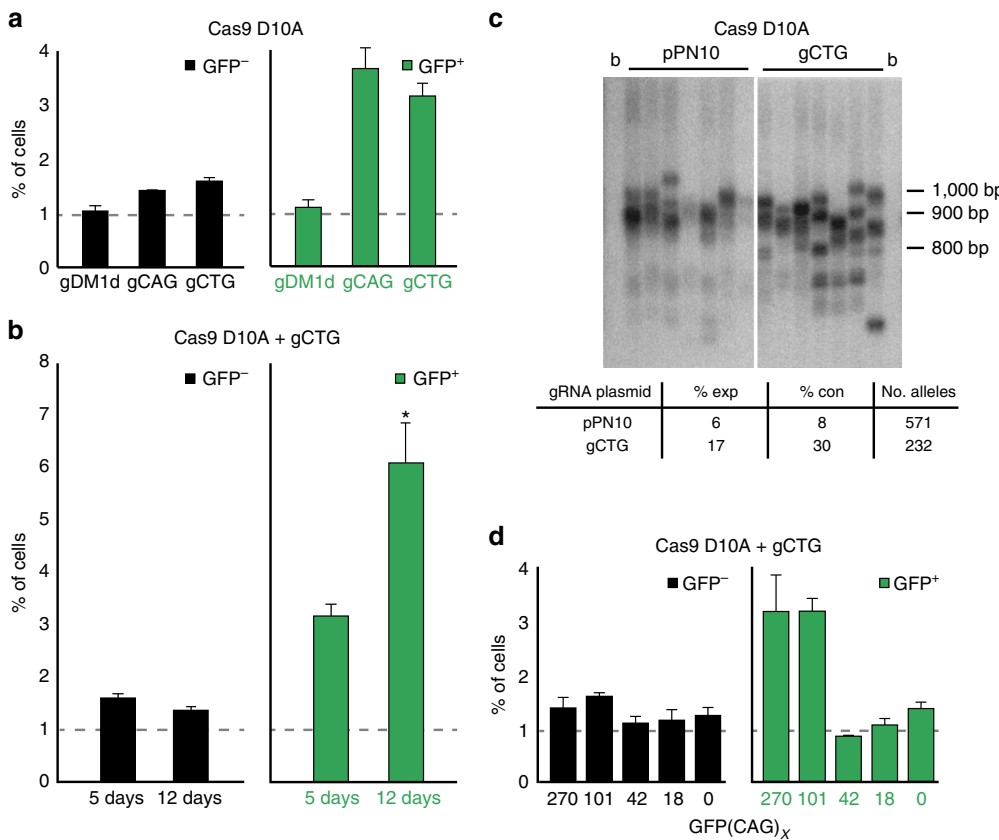

**Figure 2 | The Cas9 nickase causes CAG repeat contraction.** (**a**) Quantification of the effect of Cas9 nickase expression in GFP(CAG)$_{101}$ cells. Number of replicates per treatment: gCTG, $n = 37$; gCAG, $n = 3$; gDM1d, $n = 3$. Quantification of gCTG experiments also include results from Cas9 nickase expression treated with DMSO and siRNAs against vimentin. Dashed line: dimmest (GFP$^-$) or brightest (GFP$^+$) 1% of the cells transfected with the Cas9 nickase vector and the empty gRNA plasmid. (**b**) Quantification of GFP$^-$ and GFP$^+$ cells after 5 days (1 transfection) or 12 days (3 transfections) of expressing the Cas9 nickase along with gCTG in GFP(CAG)$_{101}$ cells. Number of replicates per treatment: 5 days, $n = 37$ (same as in **a**); 12 days, $n = 4$ (*$P = 0.001$, using a Wilcoxon $U$-test, compared with 5-day treatment). Dashed line: dimmest (GFP$^-$) or brightest (GFP$^+$) 1% of the cells transfected with the Cas9 nickase vector and the empty gRNA plasmid over the indicated period of time. (**c**) SP-PCR (top) and its quantification (bottom) of DNA isolated from GFP(CAG)$_{101}$ cells after Cas9 nickase expression with or without gCTG (shown: 100 pg DNA per PCR) or pPN10 (50 pg DNA per PCR) for 12 days. b, no DNA blank. (**d**) The ability of the Cas9 nickase to induce contractions is repeat-length dependent. Dashed line: dimmest (GFP$^-$) or brightest (GFP$^+$) 1% of the cells transfected with the Cas9 nickase vector and the empty gRNA plasmid in the indicated cell line. For each cell line the 1% threshold is determined independently. The error bars are s.e.m. Number of replicates per treatment: GFP(CAG)$_{270}$, $n = 3$; GFP(CAG)$_{101}$, $n = 37$ (same as in **a**); GFP(CAG)$_{42}$, $n = 3$; GFP(CAG)$_{18}$, $n = 2$; GFP(CAG)$_{0}$, $n = 13$.

**Table 1 | Effect of the Cas9 nickase targeted by gCTG at CAG/CTG sites in the genome.**

| Locus | No. of repeats* | | No. of alleles sequenced | No. with changes |
|---|---|---|---|---|
| | Allele 1 | Allele 2 | | |
| *AR* | 20 + 5 | 21 + 5 | 18 | 0 |
| *ATN1* | 15 | 16 | 18 | 0 |
| *ATXN1* | 12 + 11 | 12 + 12 | 18 | 0 |
| *DMPK* | 5 | 5 | 18 | 0 |
| *PPP2R2B* | 10 | 10 | 18 | 0 |
| *TBP* | 9 + 18 | 9 + 19 | 18 | 0 |
| *TCF4* | 14 | 17 | 18 | 0 |

See Supplementary Table 1 for sequence composition at these loci.
*Alleles from GFP$^+$ cells sorted from GFP(CAG)$_{101}$ cells transfected with the Cas9 nickase and gCTG expressing plasmids.

nickase and gCTG (Fig. 2c). The number of contractions accounted for nearly a third of the total alleles compared with only 8% when cells were transfected with the Cas9 nickase-expressing vector alone (Fig. 2c, $P < 0.0001$, using a Fisher's exact test). On nickase expression there was also an increase in the number of expansions, but there were fewer of them and the changes in size were smaller than for the contractions. We conclude that the Cas9 nickase targeted by gCAG or gCTG leads to a marked bias towards contractions, which is in sharp contrast to the results we obtained with the ZFN and the Cas9 nuclease.

We next examined the effect of repeat length on Cas9 nickase-induced contractions. To do so, we used GFP(CAG)$_x$ cell lines with repeat sizes ranging from 0 to 270 CAGs. We detected slight increases of 1.2- to 1.6-fold in GFP$^-$ cells on expression of both the Cas9 nickase and gCTG. This effect was largely independent of the repeat size, suggesting that this slight increase in GFP$^-$ cells seen in GFP(CAG)$_{101}$ is only partly caused by changes in repeat length (Fig. 2d). By contrast, the same treatment increased the proportion of GFP$^+$ cells in GFP(CAG)$_{270}$ and GFP(CAG)$_{101}$ cells, but not in GFP(CAG)$_{42}$, GFP(CAG)$_{18}$ nor GFP(CAG)$_0$ (Fig. 2d). These observations suggest that normal-length repeats are not prone to instability on action of the Cas9-nickase. We further substantiated this claim by examining the extent of the Cas9-induced changes at seven different loci in the genome harbouring repeats of normal sizes (Supplementary Table 2). We used nine GFP$^+$ clones with contractions within the GFP reporter caused by the action of the Cas9 nickase guided by gCTG. Of the 126 alleles sequenced, we found that they all remained mutation-free (Table 1), suggesting that the frequency of off-target mutations caused by the nickase is low. Together, these results argue that expanded CAG repeats are targets of the Cas9 nickase, leading predominantly to contractions.

**SSBR is not involved in Cas9 nickase-induced contractions.** Our results suggest that the type of damage induced within the repeat tract influences instability. It was therefore important to confirm the mutagenic intermediate created by the Cas9 nickase. The simplest hypothesis is that DNA nicks are themselves mutagenic. We therefore tested the effect of X-Ray Repair Cross-Complementing Protein 1 (XRCC1) and Poly (ADP-ribose) polymerase (PARP) activities on nickase-induced instability. The XRCC1–PARP1 complex works as a nick sensor and is involved in their repair[34]. In addition, XRCC1 interacts with a number of DNA glycosylases and is required for the repair of single-nucleotide gaps, i.e., SSBs that arise during BER[35]. This is highly relevant because BER causes expansion in a Huntington disease mouse model[17,18], and both XRCC1 and PARP1 protect against contractions in a mammalian-based assay that is blind to expansions[20]. Therefore, the prediction was that the knockdown of *XRCC1* or the inhibition of PARP using Oliparib would

significantly affect the contraction frequencies caused by the Cas9 nickase if DNA nicks or SSBs are mutagenic. This prediction was not confirmed: neither the knockdown of *XRCC1* nor the chemical inhibition of PARP activity changed the frequency of GFP$^+$ cells compared with controls (Fig. 3a,b and Supplementary Fig. 4A,B). We confirmed that the XRCC1 protein levels were substantially reduced and that the Oliparib concentration used led to an accumulation of cells in G2 and that it inhibited PARylation in response to Zeocin assault (Supplementary Table 3 and Fig. 3a,b). These observations suggest that the mutagenic intermediate is neither a DNA nick nor a SSB, and imply that nickase-induced contractions occur through a mechanism that is distinct from spontaneous and BER-dependent CAG repeat instability. We posit instead that DNA gaps larger than a single nucleotide may lead to nickase-induced contractions.

**ATR and ATM in Cas9 nickase-induced CAG repeat instability.** DNA gaps, for example, those induced by ultraviolet light during G1 of the cell cycle, activate ATR[36]. We therefore tested the effect of inhibiting this DDR kinase on nickase-induced instability using the small molecule VE-821 (ref. 37). We found that this inhibitor led to a 3.1- and 5.9-fold increase in GFP$^-$ and GFP$^+$ cells, respectively, when used in combination with the Cas9-nickase and gCTG (Fig. 4a, $P = 0.03$ compared with dimethylsulphoxide (DMSO) treated cells, using a Wilcoxon U-test). This treatment did not affect GFP expression in GFP(CAG)$_0$ (Supplementary Fig. 4B), confirming that the effect depends on the Cas9 nickase activity within the expanded repeat tract. These data suggest that ATR prevents CAG repeat instability at Cas9 nickase-induced damage.

ATM is a related DDR kinase that is partially redundant with ATR[38]. Thus, we wanted to know what effect ATM might have on nickase-induced contractions. KU60019, a specific inhibitor of ATM[39], led to a nearly two-fold reduction in the frequency of GFP$^+$ cells compared to DMSO-treated cells (Fig. 4a, $P = 0.01$, using a Wilcoxon U-test). To test whether the effect of ATR was dependent on the activity of ATM, we treated the cells with both inhibitors simultaneously. This double treatment reduced the number of contractions induced by the Cas9 nickase compared with DMSO-treated cells ($P = 0.03$, using a Wilcoxon U-test), to a level similar to using the ATM inhibitor alone (Fig. 4a, $P = 0.57$, using a Wilcoxon U-test). These observations suggest that the activity of ATM is required to cause nickase-induced instability in the absence of ATR.

**A role for MSH2 and XPA in the absence of ATR activity.** We next aimed to further define how the Cas9 nickase leads to a contraction bias. A central player in CAG repeat instability is MutS Homolog 2 (MSH2), which is essential for mismatch repair. *MSH2* knockout in mouse models and its knockdown in human cell-based assays nearly eliminates expansions[10,40,41]. Its role in

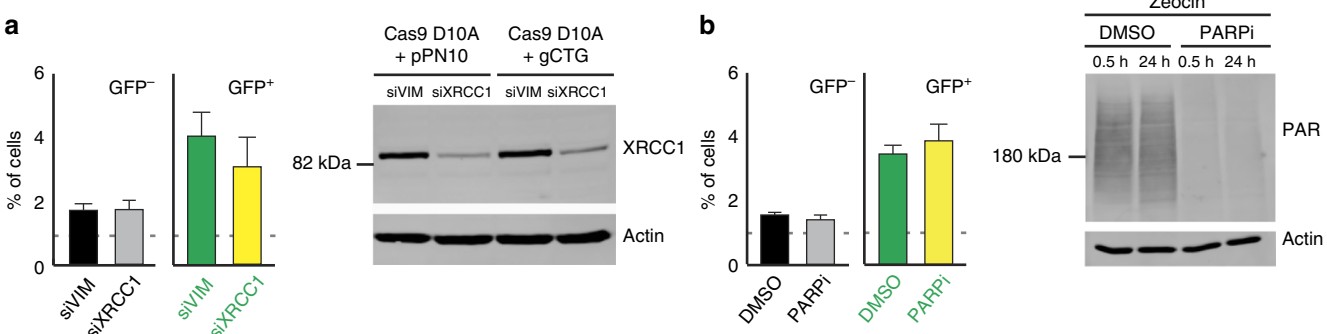

**Figure 3 | SSB repair is not involved in Cas9-nickase-induced repeat instability.** (**a**) Left: XRCC1 knockdown ($n=4$) did not affect the GFP expression in GFP(CAG)$_{101}$ cells ($P=0.7$ for both GFP$^-$ and GFP$^+$ cells, using a Wilcoxon $U$-test). Dashed line: dimmest (GFP$^-$) or brightest (GFP$^+$) 1% of the cells transfected with the Cas9 nickase vector; the empty gRNA plasmid; and the indicated siRNAs. Right: western blot showing knockdown efficiency. (**b**) Left: same as **a**, but cells treated with the PARP inhibitor Oliparib ($n=4$, $P=0.5$ for GFP$^-$ cells and $P=0.4$ for GFP$^+$ cells compared with DMSO-treated cells, using a Wilcoxon $U$-test). Dashed line: dimmest (GFP$^-$) or brightest (GFP$^+$) 1% of the cells transfected with the Cas9 nickase vector together with the empty gRNA plasmid, and treated with either DMSO or Oliparib. Right: PAR levels 30 min and 24 h after treatment with 100 μg ml$^{-1}$ of Zeocin. The error bars represent the s.e.m.

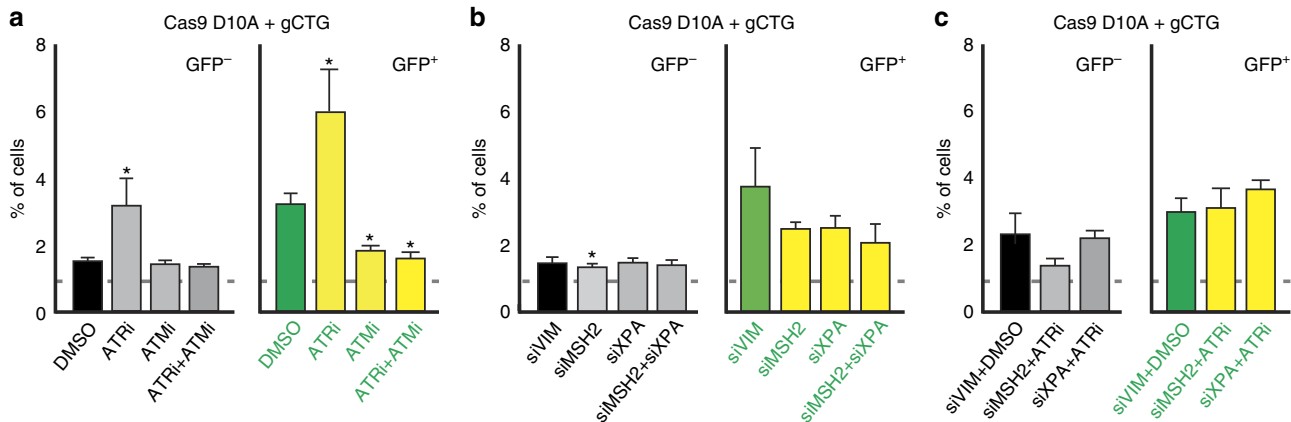

**Figure 4 | Mechanism of Cas9-nickase-induced repeat instability.** (**a**) Quantification of GFP$^-$ and GFP$^+$ cells on treatment with DMSO ($n=20$, which includes the amount of DMSO for treatment with one or two inhibitors in GFP(CAG)$_{101}$ cells; these controls were indistinguishable from each other), an ATR inhibitor (VE-821, $n=5$), or an ATM inhibitor (KU60019, $n=5$), or both ($n=3$). Dashed line: dimmest (GFP$^-$) or brightest (GFP$^+$) 1% of the cells transfected with the Cas9 nickase vector together with the empty gRNA plasmid and treated with the DMSO or the indicated inhibitor. (**b**) Quantification of the GFP$^-$ and GFP$^+$ cells on knockdown with siRNAs (siVIM, $n=13$; siMSH2, $n=14$; siXPA, $n=9$; siMSH2+siXPA, $n=4$). siVIM quantifications include results from knockdown of vimentin with both 10 and 20 nM of siRNAs as the results were indistinguishable from each other. Dashed line: dimmest (GFP$^-$) or brightest (GFP$^+$) 1% of the cells transfected with the Cas9 nickase vector; the empty gRNA plasmid; and the indicated siRNAs. (**c**) Quantification of the GFP$^-$ and GFP$^+$ cells on combinatorial knockdown of the indicated siRNAs and the ATR inhibitor VE-821 (siVIM+DMSO, $n=6$; siMSH2+ATRi, $n=3$; siXPA+ATRi, $n=6$). Dashed line: dimmest (GFP$^-$) or brightest (GFP$^+$) 1% of the cells transfected with the Cas9 nickase vector together with the empty gRNA plasmid, and treated with the indicated inhibitors and siRNAs The error bars are s.e.m. *$P\leq0.05$.

contraction, however, is more controversial. In mouse models, knocking out *MSH2* either promoted or had no effect on contractions[10]. In human cells *MSH2* downregulation promotes contractions or instability in both directions, depending on the model system used[15,40,42,43]. We found that *MSH2* knockdown did not consistently reduce the number of Cas9 nickase-induced GFP$^+$ cells compared with a control knockdown of vimentin (Fig. 4b, $P=0.14$, using a Wilcoxon $U$-test). MSH2 promotes CAG repeat contractions together with the NER factor, Xeroderma Pigmentosum, Complementation Group A (XPA)[15], in a human cell-based assay. XPA is also required for CAG repeat instability in mouse neuronal tissues[44] and for contractions in a human cell-based assay[15]. It was therefore not surprising that the knockdown of *XPA* alone or in combination with *MSH2* knockdown did not significantly reduce the frequency of nickase-induced GFP$^+$ cells (Fig. 4b, $P=0.18$, using a

Wilcoxon $U$-test, for comparing *XPA* and vimentin (*VIM*) knockdowns; and $P=0.07$, using a Wilcoxon $U$-test, when comparing double knockdown to vimentin knockdown). These results argue that neither MSH2 nor XPA are involved in generating contractions at Cas9 nickase-induced lesions.

We reasoned that ATR inhibition may be increasing the number of expansions and contractions because DSB intermediates may form under these conditions. We therefore tested whether the NER pathway, which is known to generate DSBs on ultraviolet damage[45] and at short inverted repeats[46], could contribute to repeat instability in the absence of ATR activity. Knockdown of *XPA* in cells treated with VE-821 led to results indistinguishable from those obtained when treating cells with DMSO together with a control siRNA (Fig. 4c, $P=0.70$, using a Wilcoxon $U$-test). Similarly, the effect of VE-821 treatment was suppressed by *MSH2* knockdown (Fig. 4c, $P=0.71$ compared

with control DMSO and vimentin siRNA treatments, using a Wilcoxon *U*-test). These results suggest that expansions and contractions induced by the inhibition of ATR occur because of a XPA- and MSH2-dependent activity that may eventually generates DSBs.

## Discussion

Many assays have been used with great success to dissect the mechanisms of repeat instability. Unfortunately, they are often slow, labour-intensive and/or cannot probe both expansions and contractions at once[15,24,33,47–49]. Here we have adapted a chromosomal-based reporter assay such that it can monitor instability in both directions within only 5 days. We show that the assay can be coupled to pharmaceutical treatments, siRNAs and cDNA overexpression, making it highly versatile and well suited for screening.

DNA nicks appear to be repaired by distinct and still poorly understood mechanisms. For example, they stimulate homology-directed repair in a human cell-based assay[50]. Intriguingly, this process is suppressed by RAD51 and BRCA2, which are required for homologous recombination at DSBs[50]. PARP inhibition also stimulates nick-induced homology-directed repair[51]. Our observation that the same PARP inhibitor has no effect on nickase-induced contraction is suggestive of a different pathway being used at CAG repeats and that DNA nicks are not the mutagenic intermediates leading to nickase-induced contractions. Furthermore, the lack of an effect when knocking down XRCC1 or inhibiting PARP1 implies that the Cas9 nickase leads to contraction via a pathway different from that of BER-generated SSBs. We cannot rule out that DNA nicks lead to contractions independently of the known pathways leading to spontaneous instability. Instead, however, we offer a model (Fig. 5) whereby the Cas9 nickase induces several nicks on the same strand within the repeat tract, thereby generating DNA gaps. This hypothesis is attractive because it provides an explanation for the repeat-length dependency of nickase-induced contractions: shorter repeats have fewer gCTG-binding sites and thus DNA gaps are not created as readily, leading to a stable tract. Together, these observations suggest that different types of DNA lesions found within the repeat tract are repaired by different pathways, which may dictate the direction of repeat instability.

DNA gaps are important intermediates in CAG repeat instability in model systems as varied as yeast and mice[52–54]. How they lead to contraction, however, has remained unclear. In our model (Fig. 5), we propose that DNA gaps caused by the Cas9 nickase are converted to contractions via an ATM-dependent mechanism—perhaps by promoting ligation of single-stranded DNA ends across a hairpin. This intermediate could be further processed or simply replicated in the following cell cycle to create a contraction. DNA gap filling, promoted by ATR, would prevent the involvement of ATM, providing an explanation for the apparent role of ATR in antagonizing ATM. When ATR signalling is compromised, an intermediate, possibly stabilized by MSH2 (ref. 55) and/or XPA[56], lingers and is processed more often by an XPA-dependent recruitment of downstream nucleases. The resulting DSB is further repaired via the same error-prone pathway that processes ZFN and Cas9-induced DSBs.

The yeast homologue of ATR, Mec1, prevents the appearance of contractions, most likely by preventing DSB formation at expanded CAG repeats[21]. Tel1, the ATM homologue, had no effect on CAG repeat instability[14]. Admittedly, budding yeast displays a bias towards contractions in wild-type cells and may therefore process CAG repeats differently than human cells. Nevertheless, it is unclear why the roles that we have uncovered here should be different than the ones uncovered in yeast. One

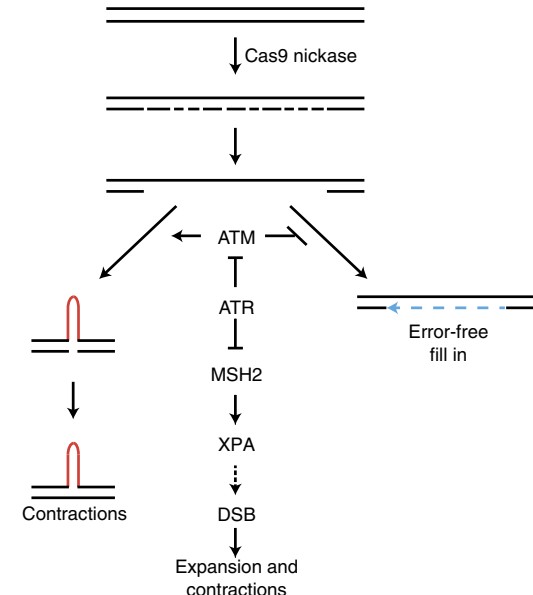

**Figure 5 | Model for Cas9-nickase-induced repeat contraction.**

possibility is that the mutagenic intermediates that Mec1 and Tel1 are sensing in the yeast studies were different than those involved here.

ATR and ATM heterozygosities also have distinct effects on the instability of CGG/CCG repeats in mice. In agreement with data presented here, ATR prevents the expansion of CGG/CCG repeats both in somatic tissues as well as in non-replicating prophase I-arrested mouse oocytes[57]. The effect of ATR on contractions was not reported. $Atm^{+/-}$ animals, by contrast, did not display an overt somatic instability phenotype. Instead, they showed markedly increased frequencies of expansions in the male germlines[58]. The effect on contractions was not reported. The reason for these differences is currently unclear but may include the very different nature of the trinucleotide repeats studied (CAG/CTG versus CGG/CCG), and/or the difference between the human cells used here and the *in vivo* mouse model used in both previous studies. More work is required to resolve this issue.

Our results have profound implications for somatic gene editing of expanded CAG diseases. Programmable nucleases were proposed to provide a tool to shorten repeat tracts and a much needed cure[59]. Some attempts have been made to test this hypothesis using ZFNs or TALENs[24,26,60,61]. Our data caution that inducing DSBs within the repeat tract in an attempt to shrink them would also lead to repeat expansion. This would be a problem because the expansions are likely to exacerbate the disease phenotype[22,23]. An alternative may be to induce two DSBs in regions immediately flanking, but not within, the repeat tract. This approach would be prone to off target effects[62–64], may lead to the mutation of the wild-type allele, and gRNAs would have to be designed and tested for each disease locus. Our approach would be simpler, using a single gRNA that could target any of the disease loci. In addition, our data argue that only longer, pathogenic, repeat tracts are targeted for contractions; an ideal scenario as it leaves the normal allele intact.

For CAG repeat contraction to be a viable therapeutic avenue, the disease phenotypes must be reversible. There is some evidence that this is the case. Indeed, the myotonia and cardiac symptoms of a myotonic dystrophy mouse model were reversible on shutting off the expression of the pathogenic transgene[65]. Similarly, halting the expression of a spinocerebellar ataxia type 1 allele with 82 CAGs markedly improved the pathological

phenotype of Purkinje cells and reversed motor dysfunction[66]. Homology-directed replacement of an expanded CAG repeat in induced pluripotent stem cells (iPSC) derived from Huntington disease patients improved suceptibility to cell death and mitochondrial defects[67]. Removing a CGG/CCG repeat tract along with flanking sequences from the *FMR1* gene with CRISPR-Cas9 nuclease reactivated the expression of FMRP in a few iPSC clones[68]. Finally, excising expanded GAA/TTC repeats in Friedreich Ataxia fibroblasts reactivated the expression of frataxin, improved the activity of the Fe-S-containing Aconitase, and increased cellular ATP levels[69]. Together with our results, these studies offer great hope that Cas9 nickase-mediated shrinkage of expanded repeat tracts in somatic tissues may alleviate disease symptoms in patients.

## Methods

**Cell culture.** The GFP(CAG)$_0$ and GFP(CAG)$_{101}$ cells lines were a kind gift from John H. Wilson[24]. The cells tested negative for mycoplasma using the MycoAlert detection kit (Lonza) at the start of our experiments and during the revisions of this manuscript. The GFP(CAG)$_{15}$, GFP(CAG)$_{18}$, GFP(CAG)$_{42}$, GFP(CAG)$_{50}$ and GFP(CAG)$_{270}$ were isolated from populations grown for 6 months unperturbed or after transfection with the ZFN. They did not contain mutations in the region flanking the repeat tract. The cells were maintained at 37 °C with 5% $CO_2$ in Dulbecco's modified Eagle's medium (DMEM) glutamax, supplemented with 10% fetal bovine serum (FBS), 100 U ml$^{-1}$ penicillin (pen), 100 µg ml$^{-1}$ streptomycin (strep), 15 µg ml$^{-1}$ blasticidine and 150 µg ml$^{-1}$ hygromycin. When the cells were destined for flow cytometry, they were kept in DMEM glutamax, with 10% of dialysed calf serum, along with pen–strep. During the long-term culturing, the unstransfected and unperturbed cells were split one to five twice a week, and the medium was supplemented with blasticidine and hygromycin to ensure continued expression of the TetR and GFP transgenes.

**Plasmids and siRNA transfections.** The plasmids used in this study are found in Supplementary Table 4. They are available on request. cDNA transfections were performed using $6 \times 10^5$ cells per well in 12-well plates using a total of 1 µg of DNA and Lipofectamine 2000 (Life Technologies) per well. The culture medium was replaced 6 h after transfection and 2 µg ml$^{-1}$ of dox, diluted in DMSO, was added. Controls without dox were treated with DMSO alone. Forty-eight hours later, the medium was replaced and dox was freshly added. Flow cytometry, protein extraction and/or DNA extraction were performed after another 48 h of incubation.

The siRNAs used in this study are found in Supplementary Table 5. When transfecting with both a cDNA and a siRNA, $8 \times 10^5$ cells per well were used along with 1 µg of DNA and 20 nM of siRNAs using Lipofectamine 2000. The medium was replaced 6 h later and dox was added. Forty-eight hours after the first transfection, we performed a second siRNA transfection with RNAiMax (Life Technologies) using half of the cells present and 20 nM of siRNA. We collected the cells to assess knockdown efficiency or GFP fluorescence analysis 48 h later. When transfecting two siRNAs, we used a final siRNA concentration of 40 nM, where 20 nM of each individual siRNA were used. We found that single knockdowns at 20 nM were no different from those also containing 20 nM of the vimentin siRNA and were pooled for the statistical analyses and in the presented figures.

**Pharmacological inhibitors.** When using small-molecule inhibitors (Supplementary Table 6), the cells were treated as above. The medium, along with the dox and the inhibitors, was replaced after 48 h and for another 48 h of treatment. Cell cycle analysis was performed after 96 h of treatment. Briefly, the cells were fixed with 100% ethanol and treated with RNAseA (50 µg ml$^{-1}$) before adding propidium iodine (50 µg ml$^{-1}$). Flow cytometry analysis was performed as described below.

**Flow cytometer and cell sorting.** In preparation for flow cytometry analysis, cells were re-suspended in phosphate-buffered saline (PBS) with 1 mM EDTA to a concentration of about $10^6$ cells per ml. For each condition, we measured at least $2 \times 10^5$ events using a LSRII from BD. Data analysis was done using Flowing II. FACS was performed using a FACS Aria II (BD) or MoFlo Astrios (Beckman Coulter). For single-clone analyses, we re-suspended the cells to a concentration of $2 \times 10^6$ cells per ml and sorted the GFP$^-$ and GFP$^+$ cells. The cells were then expanded in DMEM glutamax supplemented with pen–strep, blasticidine, hygromycin, 5% FBS and 5% dialysed calf serum. For viability tests, cells were treated as described above except that 96 h after the first transfection they were collected in PBS with 1 mM EDTA, and 1 µM of TO-PRO-3 was added as a dead cell marker.

**Quantification of GFP$^-$ and GFP$^+$ cells.** To quantify the fold increase in the number of GFP$^-$ or GFP$^+$ cells, we first established gates that contained the top or bottom 1% of GFP-expressing cells in the control treatment, for example, the nickase plasmid transfected together with an empty gRNA vector (pPN10). For each treatment or cell line, therefore, the top and bottom 1% were adjusted to take

any shift in GFP expression into account. In some cases, we adjusted the voltage of the flow cytometer laser to accommodate samples with very high or very low GFP expression. This adjustment did not interfere with the quantification (Supplementary Fig. 3C,D). Once the GFP gates were established, we calculated the percentage of cells from the test population (for example, expressing both the Cas9 nickase and the gCTG) falling within these same gates. In cases where inhibitors or siRNAs were used, the control population expressed the Cas9 nickase, pPN10 and the inhibitor or siRNA. The 1% cutoffs were used to keep a balance between having enough cells for robust statistics and detecting significant fold changes[24]. This method probably underestimates the frequencies of change compared with SP-PCR (Fig. 2).

**Repeat length determination and SP-PCR.** To determine the repeat length of each sorted clone, we isolated DNA using the PeqGold MicroSpin Tissue DNA kit (PeqLab). The DNA was then amplified with primers oVIN-0437 and oVIN-0459 (Supplementary Table 7). Several PCR reactions were set-up with MangoTaq and the products were gel-extracted, pooled and sent for sequencing with the same primers used for the amplification. The repeat size was determined from at least two different amplification and sequencing reactions. The longest repeat size determined was used in the rare cases where the repeat length was not identical between the runs. SP-PCR was done based on the protocol described in ref. 70. Briefly, primers oVIN-0459 and oVIN-0460 were used for the amplification along with between 50 and 100 pg of genomic DNA per PCR. The products were then run on an agarose gel and transferred into a membrane. The probe was derived from a PCR product amplified with the same primers from a plasmid containing 40 repeats. The primers used to amplify the off-target loci are found in Supplementary Table 7.

**Antibodies and western blotting.** Protein extraction was done using RIPA buffer and proteinase inhibitor cocktail tablets (Roche, Germany) and at least 10 µg of proteins were loaded onto 6 or 10% Tris/glycine SDS polyacrylamide gels and transferred onto nitrocellulose membranes. The antibodies used in this study are found in Supplementary Table 8. An Odyssey Infrared Imager (Licor) was used for signal detection. All uncropped western blots are found in Supplementary Fig. 5.

**Statistics.** When determining whether there were differences in the frequency of GFP$^-$ and GFP$^+$ cells between treatments, we were unable to guarantee that the data were normally distributed using a two-tailed Kolmogorov–Smirnov test. We therefore used a two-tailed Wilcoxon *U*-test as it is non-parametric. We also performed two-tailed Student's *t*-tests, which gave similar results as the *U*-tests. The same was true when comparing length of the repeat tracts in clones sorted from different populations. We used a Poisson distribution to evaluate the total number of alleles amplified in our SP-PCR experiments based on the proportion of PCRs that did not yield a detectable product. Fisher's exact tests were used to determine whether there were changes in the number of contractions and expansions seen in the SP-PCR experiment. All statistical analyses were done using R Studio version 0.99.441. We concluded that a significant difference existed when $P < 0.05$.

**Data availability.** The data presented in this study are available from the corresponding author.

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

## Acknowledgements

We thank F. Hamaratoglu, J.H. Wilson and members of the Dion laboratory for helpful discussions. H. Ferreira, G. Gourdon, F. Hamaratoglu, J. E. Martin, M. Op, A. Orioli and G.-F. Richard critically read the manuscript. We thank J.H. Wilson and J. Lingner for reagents; P. Nunes and A. Feola for cloning some of the gRNA vectors; and O. Rodriguez Lima for blindly quantifying the SP-PCR blots. This work is supported by a Swiss National Science Foundation professorship (#144789) and by Gebert Rüf Stiftung and UNISCIENTIA STIFTUNG within the programme «Rare Diseases – New Approaches» (GRS-060/14) to V.D.

## Author contributions

Everyone contributed to performing the experiments and analysing the results. C.C. and V.D. designed the experiments and wrote the paper.

## Additional information

**Competing financial interests:** Procedure ongoing for European Patent application number EP16165203.7.

**How to cite this article**: Cinesi, C. *et al.* Contracting CAG/CTG repeats using the CRISPR-Cas9 nickase. *Nat. Commun.* **7,** 13272 doi: 10.1038/ncomms13272 (2016).

