## [Peer Review File · Nature Communications]

Reviewer #1 (Remarks to the Author):

The revision submitted by the authors is much improved in presentation, and better integrates their DNA repair results with the assay. The authors have also included more controls, which strengthen the data. The main points: (1) a new assay system for evaluating triplet expansion where both expansions and contractions can be measured together. (2) Using the assay, the authors probe the mechanisms for expansion and deletion. The authors find that expansion occurs during DSB in an ATM-dependent fashion, that SSB cause contractions, and that ATR inhibition increases both expansions and contractions in an Msh2 and Xpa-dependent manner.

Despite the improvement, there remain some of the same weaknesses from the last submission. The new assay does not extend the capability of the original Wilson experiments sufficiently, and there are technical issues with the assay. Some of the results differ in unexpected ways from some generally agreed upon effects.

Technical.

One of the major points of the manuscript is the generation of a new assay method, but I have technical concerns about (1) the size overlap in the GFP- and GFP+ pools, (2) whether there is really selection bias for expansions, and (3) the small size of the repeat changes.

(1) There is no selectable marker in the strict sense, but is based on GFP intensity. This is a reasonable selection method. However, in this assay, it is a concern because, even with GFP selection, clones both in the top and the bottom 1% of GFP intensity (GFP+ and GFP-) thresholds contained both expansions and contractions (Fig. S1E.F). The authors depend on a small number of clones which are outliers to report the impact of repair factors, but since the two groups overlap in size, the results may not be reliable.

(2) The authors point is that expansion and contraction bias can be measured in the same assay, but there does not seem to be a significant expansion bias. Indeed, the authors present data on a larger set 232 alleles (Fig. 2E), but when all them are considered, the number of expansions and contraction were not really "biased": expansions were 7% and contractions were 12%. Given that GFP- and GFP+ pools partially overlap, perhaps it would be better to plot the results as a box plot to see how significantly different are the distributions.

(3) Except for 3 clones, the changes in length of the tract sizes are small, and therefore unavoidably variable. The impact of repair on the direction bias of the change may be difficult to control for unless hundreds of clones are assayed. However, increasing the pool size is not really possible since the GFP- and GFP+ pools become increasingly mixed and more similar in size.

(4) The results of DNA factor expression and the meaning of the changes in transiently transfected cells is difficult. The authors grow cells in culture for 5-12 days and sometime 6 months to assess repeat tract length, and there is an assumption that the conditions are constant. However, after 4 days in culture, plasmids will be ejected from the cell and degraded, or will be stably incorporated. This may change the pattern of instability in ways that will be difficult to control for.

(5) GFP- clones increase their CAG length with 6 months of dox treatment -, but so do the GFP+ clones (Fig S1C and S1F). The number should decrease if the thesis of the author's is correct. For the GFP- cells, only three-four clones are shown with large expansions.

Confusing points and overstatements.

- The authors should explain why there are ATM and ATR differences or what this could mean. ATM

phosphorylates targets leading to cell cycle arrest, DNA repair, or apoptosis, while ATR primarily responds to stalled replication forks—are these differences tested by the assay system? If so, how? The statement that one occurs after the other is unsubstantiated.

- What does it mean that ATM inhibition increases instability in an XPA- and MSH2-dependent manner? Do the authors mean that MSH2 (mismatch) and XPA pathways (NER or TCR) do not rescue the DSBs? Or do the authors mean that inhibition of one pathway utilizes another DSB repair process? This is not clear.
- XPA is referred to as NER, but it is also part of the TCR pathway. That instability is influenced by dox implies that TCR might be involved, but there is no testing of that pathway.
- A major conclusion of the authors is the Cas9 nickase. The author's state: "We found that inducing double-strand breaks within the repeat tract caused instability in both directions, whereas the CRISPR-Cas9 nickase induced a marked bias towards contractions....contractions most likely arose from DNA gap intermediates - rather than via single-strand break repair "

The authors do not demonstrate that the Cas9^{10A} mutant forms a gap on the repeats. The assumption needs to be substantiated.

- The author's state " We detected a slight increase of 1.2 to 1.6 fold in GFP- cells upon expression of both the Cas9 nickase and gCTG. This effect was largely independent of the repeat size, suggesting that this slight increase in GFP cells was only partly caused by changes in repeat length (Fig. 2C)"

Doesn't this argue against the hypothesis that SSB causes contraction?

Inaccuracies and novelty

(1) The role of MSH2 in causing expansion is generally agreed upon. Multiple laboratories: Wheeler, Bates, van der Boek, Messer, Usdin and others

(2) Kovtun et al. (2001) showed that expansions depended on MSH2, not deletions as indicated by the authors. This should be reported accurately.

(3) Two groups have already reported that deletions were independent of MSH2; Guordon et al. (2003) reported that CTG contractions were independent of MSH2 in a DM1 model. Kovtun et al. (2004) reported that CTG contractions were independent of MSH2 in an HD model.

- Savouret C, Brisson E, Essers J, Kanaar R, Pastink A, te Riele H, Junien C, Gourdon G. CTG repeat instability and size variation timing in DNA repair-deficient mice. *EMBO J.* 2003 22(9):2264-73.
- Kovtun I, Thornhill AR, and McMurray CT. (2004). Somatic Deletion Events Occur During Early Embryonic Development and Modify the Extent of CAG Expansion in Subsequent Generations.

(4) A third report indicates that deletions are independent of MSH3.

o Slean MM, Panigrahi GB, Castel AL, Pearson AB, Tomkinson AE, Pearson CE. Absence of MutS β leads to the formation of slipped-DNA for CTG/CAG contractions at primate replication forks. *DNA Repair (Amst).* 2016 ;42:107-

The authors report that Cas9 mutant nickase enhances SSB and contractions of the top 1% of GFP+ clones. Wilson showed, using the same kind of assay, that contractions of CAG repeats are reduced by siRNA knockdown of ERCC1, XPG and CSB, suggesting that SSB mechanisms of BER

and TCR have a causative influence on contractions in their assay.

Inconsistencies from the literature

Double-strand breaks within the repeat tract caused instability in both directions. The assay does not clarify or move the field forward. In fact, current reported suggest that DSB cause contraction and or stops instability. The use of Cas9 consistently results in deletion or certainly has a strong deletion bias.

(1) Ye Y, Kirkham-McCarthy L, Lahue RS. The *Saccharomyces cerevisiae* Mre11-Rad50-Xrs2 complex promotes trinucleotide repeat expansions independently of homologous recombination. *DNA Repair (Amst)*. 2016 Jul;43:1-8.

(2) Park CY, Halevy T, Lee DR, Sung JJ, Lee JS, Yanuka O, Benvenisty N, Kim DW. Reversion of FMR1 Methylation and Silencing by Editing the Triplet Repeats in Fragile X iPSC-Derived Neurons. *Cell Rep*. 2015 Oct 13;13(2):234-41.

(3) Sundararajan R, Gellon L, Zunder RM, Freudenreich CH. Sundararajan R, Gellon L, Zunder RM, Freudenreich CH. Double-strand break repair pathways protect against (CAG/CTG repeat expansions, contractions and repeat-mediated chromosomal fragility in *Saccharomyces cerevisiae*. *Genetics*. 201184(1):65-77).

The conclusions that SSB breaks primarily lead to deletions contradict results of many other DNA repair investigators. In mice, loss of XPA and BER enzyme suppress expansions, i.e., indicating that these SSB pathways cause expansion, and there is no evidence that these proteins are involved in DSBs.

(1) Møllersen L, Rowe AD, Illuzzi JL, Hildrestrand GA, Gerhold KJ, Tveterås L, Bjølgerud A, Wilson DM 3rd, Bjørås M, Klungland A. Neil1 is a genetic modifier of somatic and germline CAG trinucleotide repeat instability in R6/1 mice. *Hum Mol Genet*. (2012) 21(22):4939-47 (2012).

(2) Kovtun IV, Liu Y, Bjoras M, Klungland A, Wilson SH, McMurray CT, OGG1 initiates age-dependent CAG trinucleotide expansion in somatic cells. *Nature*, 447(7143): p. 447-52 (2007).

(3) Hubert L Jr, Lin Y, Dion V, Wilson JH. Xpa deficiency reduces CAG trinucleotide repeat instability in neuronal tissues in a mouse model of SCA1. *Hum Mol Genet*. 20(24):4822-3 (2011).

The authors may be correct, but a more in depth analysis beyond the assay needs to be presented to support a divergent point of view or strengthen their conclusions.

Reviewer #2 (Remarks to the Author):

The authors have adequately addressed the majority of my concerns raised in the original critique of the manuscript. There are just a few minor points / concerns that remain.

1) The PAM sequences for the CAG/CTG repeat targets are not listed in Table S2 as indicated in the authors' response to the reviewers.

2) Given that a suboptimal PAM sequence is being employed for these nucleases/nickases, it is not surprising that the fraction of affected cells (e.g. GFP+) is modest. However for the 12 day treatment shown in Figure 2C where there is a 6% shift in the number of cells in the highest GFP gate but little increase in expansions, I would anticipate that the distribution of the entire population of cells would be shifted toward GFP+. It would be helpful to include the Flow population data for this experiment relative to a 12 day control in the supplement for readers to be able to assess the raw data as in Figure S2A.

3) The Figure 2 citations on page 6 need to be adjusted to point to the correct panel. (Fig 2E not 2C)

Reviewer #1

The revision submitted by the authors is much improved in presentation, and better integrates their DNA repair results with the assay. The authors have also included more controls, which strengthen the data. The main points: (1) a new assay system for evaluating triplet expansion where both expansions and contractions can be measured together. (2) Using the assay, the authors probe the mechanisms for expansion and deletion. The authors find that expansion occurs during DSB in an ATM-dependent fashion, that SSB cause contractions, and that ATR inhibition increases both expansions and contractions in an Msh2 and Xpa-dependent manner.

We thank the reviewer for highlighting the improvements and for her/his comments, which have very much helped us to bring the manuscript up to this level.

Despite the improvement, there remain some of the same weaknesses from the last submission. The new assay does not extend the capability of the original Wilson experiments sufficiently, and there are technical issues with the assay. Some of the results differ in unexpected ways from some generally agreed upon effects.

We believe that adapting the original GFP assay published by the Wilson lab is a significant leap forward even if the technical adjustments were small. This is because having a chromosomal reporter assay that measures CAG repeat instability in both directions within 5 days is invaluable to screen treatments quickly for biases in instability. Such an assay has been sought after ever since the first chromosomal reporter was limited to measuring rare large contractions (Gorbunova et al Mol Cell Biol. 2003 Jul;23(13):4485-93).

We validated the accuracy of our measurements by:

- 1) reproducing the results of Santillan et al (Fig. 1B).
- 2) sequencing single GFP⁻ and GFP⁺ clones, supported by new statistics (Fig.S1, S2, S3).
- 3) by small-pool PCR of sorted GFP⁻ and GFP⁺ sorted cells (see below).
- 4) confirming that the Cas9 nickase induces larger and more frequent contractions in an assay that is completely independent of the GFP reporter, namely small-pool PCR on the unsorted, bulk population (Fig. 2C).

Given that all the tests of the assay agree with each other and with published literature, we conclude that our assay robustly detects changes in repeat length in both directions.

In addition, we agree with the reviewer that many of the mechanistic details that we have uncovered with the Cas9 nickase are not consistent with what is known about spontaneously occurring or BER-dependent instability. We conclude from these differences that the mutagenic intermediates induced by the Cas9 nickase are unlikely to be single-strand breaks (SSBs) and we propose instead a model in which DNA gaps play that role.

Technical.

One of the major points of the manuscript is the generation of a new assay method, but I have technical concerns about (1) the size overlap in the GFP⁻ and GFP⁺ pools, (2) whether there is really selection bias for expansions, and (3) the small size of the repeat changes.

(1) There is no selectable marker in the strict sense, but is based on GFP intensity. This is a reasonable selection method. However, in this assay, it is a concern because, even with GFP selection, clones both in the top and the bottom 1% of GFP intensity (GFP⁺ and GFP⁻) thresholds contained both expansions and contractions (Fig. S1E.F). The authors depend on a small number of clones which are outliers to

report the impact of repair factors, but since the two groups overlap in size, the results may not be reliable.

It is true that a minority of the GFP⁻ clones (12/62 or 19%) and of the GFP⁺ cells (8/91 or 9%) have changes in GFP levels that are not consistent with the size of the repeat tract. The central question is whether these populations are different from each other enough to effectively reflect changes in repeat sizes. To address this, we statistically tested whether the differences between the size of the repeat tracts between GFP⁻ and GFP⁺ clones are different from each other. We performed a two-tailed Wilcoxon U-test as it is non-parametric test. The P-values that we obtained are the following:

		GFP ⁻ vs GFP ⁺ clones	
Treatment	Figure	Wilcoxon signed-rank test	
Untreated	S1C	1x10 ⁻⁵	
6 months in DOX	S1E	0.025	
6 months in DMSO	S1G	0.035	
ZFNs	S2D	5x10 ⁻⁴	
Cas9 D10A+gCTG	S3E	1.8x10 ⁻⁴	
Cas9 D10A+gCAG	S3G	1.5x10 ⁻⁶	

We also performed the same analysis using a Student's T-test and the results agreed: the size of the repeat tracts in GFP⁻ and GFP⁺ cells are significantly different every one of the 6 times that we repeated this experiment. These results demonstrate that we analyzed enough clones to draw reliable conclusions. The implication is that the assay is accurate in detecting changes in repeat length.

To test this point further and avoid potential skewing of the data because of the isolation and growth of single clones, we also performed small-pool PCR on the sorted GFP⁻ and GFP⁺ cells after transfection with the nickase together with the gCTG. As the figure on the right shows, the size of the repeat tracts in the sorted cells is very different between the two populations, which is consistent with the data obtained from the sorted clones. This further supports the validity and robustness of the GFP-based assay.

(2) The authors point is that expansion and contraction bias can be measured in the same assay, but there does not seem to be a significant expansion bias. Indeed, the authors present data on a larger set 232 alleles (Fig. 2E), but when all them are considered, the number of expansions and contraction were not really "biased": expansions were 7% and contractions were 12%. Given that GFP⁻ and GFP⁺ pools partially overlap, perhaps it would be better to plot the results as a box plot to see how significantly different are the distributions.

The reviewer is correct in pointing out that there is no bias in the control population (transfected with the Cas9 nickase but without the gRNA) as measured by a GFP-independent assay, small-pool PCR (6% expansions vs 8% contractions) (Fig2C). We apologize if this was not clear in the text: the bias in contraction is only seen when we used the Cas9 nickase together with a gRNA targeting the repeat tract. This is apparent from the GFP-based assays in Fig. 2ABD and from the small-pool PCR analysis in Fig. 2C.

(3) Except for 3 clones, the changes in length of the tract sizes are small, and therefore unavoidably variable. The impact of repair on the direction bias of the change may be difficult to control for unless hundreds of clones are assayed. However, increasing the pool size is not really possible since the GFP⁻ and GFP⁺ pools become increasingly mixed and more similar in size.

The statistics presented in point 1 above shows that we were able to assay a sufficient number of clones and that GFP⁻ and GFP⁺ cells harbour repeat sizes that are significantly different from each other.

The changes that we detected are neither small nor variable. For instance, taking into account all the GFP⁻ cells with expansions (n=50) and all the GFP⁺ cells with contractions (n=83) that we obtained from all 6 sorting experiments (Fig. S1CEG, Fig. S2D, Fig. S3EG), a vast majority of them (74% of GFP⁻ and 96% of GFP⁺ cells) have changes of at least 10 repeats compared to the starting population. In addition, sequencing the repeat tract is very robust and we never see discrepancies of more than 1 CAG when sequencing the same sample multiple times (see methods – an issue also discussed in Dion et al HMG 2008), confirming that the assay reliably detects expansions and contractions.

(4) The results of DNA factor expression and the meaning of the changes in transiently transfected cells is difficult. The authors grow cells in culture for 5-12 days and sometime 6 months to assess repeat tract length, and there is an assumption that the conditions are constant. However, after 4 days in culture, plasmids will be ejected from the cell and degraded, or will be stably incorporated. This may change the pattern of instability in ways that will be difficult to control for.

We have now clarified in the methods that during the long term treatments, there is no transfection of plasmids or siRNA knockdowns. There is therefore no issue for these experiments. The methods contain details of how we performed the 12-day treatments found in Fig. 2B: we added two more transfections of the plasmids compared to our standard 5-day treatment. This keeps the expression of the Cas9 nickase of the gRNA high for the entirety of the experiment. During the remaining experiments using our standard 5-day regimen, we found that the Cas9 variants are expressed through the end of the experiment (Fig. S3B). In addition, knockdown of repair proteins are evident also at the end of the experiments (Fig. 3A, S4C). We believe that these controls show that protein levels are maintained throughout our experiments.

(5) GFP⁻ clones increase their CAG length with 6 months of dox treatment -, but so do the GFP⁺ clones (Fig S1C and S1F). The number should decrease if the thesis of the author's is correct. For the GFP⁻ cells, only three-four clones are shown with large expansions.

We did not have any expectations regarding the direction of instability and we apologize if our wording has led to this misconception. The main point we were trying to make with this experiment was that we could isolate expansions from GFP⁻ cells and contractions from GFP⁺ cells. And, as shown in Fig. S1E and

S1G in our manuscript, indeed we could.

Confusing points and overstatements.

- The authors should explain why there are ATM and ATR differences or what this could mean. ATM phosphorylates targets leading to cell cycle arrest, DNA repair, or apoptosis, while ATR primarily responds to stalled replication forks—are these difference tested by the assay system? If so, how? The statement that one occurs after the other is unsubstantiated.

The reason why we concluded that ATR works upstream of ATM is because of the data presented in Fig. 4A. We have now changed the wording so that what we mean is clearer. To resolve this issue, we no longer refer to ATM and ATR to be downstream or upstream of each other. Nevertheless, our data are most consistent with a genetic model whereby ATR prevents nickase-induced contractions in the same pathway as ATM. This is because if the two were in parallel pathways, then the double inhibition would lead to an intermediate phenotype. In addition, the data are not consistent with ATM being upstream of ATR because then we would have expected that the effect of ATM inhibition to be dependent on ATR inhibition, not the other way around as we observed here. Our interpretation is therefore valid.

Given the overwhelming consensus that ATM and ATR are partially redundant, we find it unlikely that ATR would inhibit ATM in a biochemical sense, for instance through phosphorylating each other or via protein-protein interaction. Instead, we imagine that ATR would promote the formation of a DNA structure that does not activate ATM. We have now updated our discussion, which reads: “DNA gaps are important intermediates in CAG repeat instability in model systems as varied as yeast and mice^{42,61,62}. How they lead to contraction, however, has remained unclear. We propose a model (Fig. 5) whereby gaps caused by the Cas9 nickase are converted to contractions via an ATM-dependent mechanism – perhaps by promoting ligation of ssDNA ends across a hairpin. This intermediate could be further processed or simply replicated in the following cell cycle to create a contraction. DNA gap filling, promoted by ATR, would prevent the involvement of ATM, providing an explanation for the apparent role of ATR in the inhibition of ATM. When ATR signaling is compromised, an intermediate, possibly stabilized by MSH2⁶³ and/or XPA⁶⁴, lingers and is processed more often by an XPA-dependent recruitment of downstream nucleases. The resulting DSB is further repaired via the same error-prone pathway that processes ZFN and Cas9-induced DSBs.”

- What does it mean that ATM inhibition increases instability in an XPA- and MSH2-dependent manner? Do the authors mean that MSH2 (mismatch) and XPA pathways (NER or TCR) do not rescue the DSBs? Or do the authors mean that inhibition of one pathway utilizes another DSB repair process? This is not clear.

We favor a model, as laid out in the discussion, whereby in the absence of ATR activity, XPA and MSH2 might stabilize the repeat tract and favor the recruitment of structure-specific nucleases. Future plans include testing this hypothesis directly to identify the potential nucleases involved.

- XPA is referred to as NER, but it is also part of the TCR pathway. That instability is influenced by dox implies that TCR might be involved, but there is no testing of that pathway.

NER is indeed divided into global-genome repair and TCR. Xpa is downstream of the merge between the two pathways. Although this is a highly interesting idea, we did not show or made any claim about how transcription might affect nickase-dependent contractions. Consequently, we did not include a discussion of this point here.

- A major conclusion of the authors is the Cas9 nickase. The author's state: "We found that inducing

double-strand breaks within the repeat tract caused instability in both directions, whereas the CRISPR-Cas9 nickase induced a marked bias towards contractions....contractions most likely arose from DNA gap intermediates - rather than via single-strand break repair "

The authors do not demonstrate that the Cas910A mutant forms a gap on the repeats. The assumption needs to be substantiated.

This is correct and we have now changed the phrasing to take into account this comment and to shorten the abstract. This passage now reads: "Using a GFP-based chromosomal reporter that monitors expansions and contractions in the same cell population, we found that inducing double-strand breaks within the repeat tract caused instability in both directions. In contrast, the CRISPR-Cas9 D10A nickase induced mainly contractions independently of single-strand break repair. Nickase-induced contractions depended upon the DNA damage response kinase ATM whereas ATR inhibition increased both expansions and contractions in a MSH2- and XPA-dependent manner. We propose that DNA gaps lead to contractions and that the type of DNA damage present within the repeat tract dictates the levels and the direction of CAG repeat instability."

Being able to detect DNA gaps directly at a specific site would be a fantastic experiment. At the moment, sadly, such an assay does not exist. We have started developing one, but this will have to wait for a future publication.

- The author's state " We detected a slight increase of 1.2 to 1.6 fold in GFP- cells upon expression of both the Cas9 nickase and gCTG. This effect was largely independent of the repeat size, suggesting that this slight increase in GFP cells was only partly caused by changes in repeat length (Fig. 2C)"

Doesn't this argue against the hypothesis that SSB causes contraction?

This statement referred to GFP⁻ cells, and thus presumably to expansions. But we do agree with the reviewer that our data argue against SSBs causing contractions. This is part of the reason why we proposed that DNA gaps might be a likelier mutagenic intermediates.

Inaccuracies and novelty

(1) The role of MSH2 in causing expansion is generally agreed upon. Multiple laboratories: Wheeler, Bates, van der Boek, Messer, Usdin and others

We have now corrected this in the text and added a number of new references. We did not include the reference from the Bates lab (presumably Gonitel et al PNAS 2008) because that reference does not test whether MSH2 promotes repeat instability. Instead, the authors argue that the levels of MSH3 correlate with repeat instability in different brain regions. We also did not add the studies from the Usdin and Pook labs on the role of Msh2 in CGG and GAA repeat instability in mice because they used different repeat tracts as the ones we are working with. He have, however, also added studies done in hESCs and human iPSCs with expanded CAG repeats that also support this conclusion.

(2) Kovtun et al. (2001) showed that expansions depended on MSH2, not deletions as indicated by the authors. This should be reported accurately.

Thank you for pointing this out. It is now corrected.

(3) Two groups have already reported that deletions were independent of MSH2; Guordon et al. (2003) reported that CTG contractions were independent of MSH2 in a DM1 model. Kovtun et al. (2004) reported that CTG contractions were independent of MSH2 in an HD model.

- Savouret C, Brisson E, Essers J, Kanaar R, Pastink A, te Riele H, Junien C, Gourdon G. CTG repeat instability and size variation timing in DNA repair-deficient mice. *EMBO J.* 2003 22(9):2264-73.
 - Kovtun I, Thornhill AR, and McMurray CT. (2004). Somatic Deletion Events Occur During Early Embryonic Development and Modify the Extent of CAG Expansion in Subsequent Generations.
- These references were added to the text and the discussion of the role of MSH2 was updated to reflect the corrections.

(4) A third report indicates that deletions are independent of MSH3.

o Slean MM, Panigrahi GB, Castel AL, Pearson AB, Tomkinson AE, Pearson CE. Absence of MutS β leads to the formation of slipped-DNA for CTG/CAG contractions at primate replication forks. *DNA Repair (Amst).* 2016 ;42:107-

We have included this reference along with a discussion of how MutS β function might explain some of the differences in the literature with respect to the role of MSH2 in generating contractions.

The authors report that Cas9 mutant nickase enhances SSB and contractions of the top 1% of GFP+ clones. Wilson showed, using the same kind of assay, that contractions of CAG repeats are reduced by siRNA knockdown of ERCC1, XPG and CSB, suggesting that SSB mechanisms of BER and TCR have a causative influence on contractions in their assay.

More precisely, we argue that the Cas9 nickase induces several nicks along the same strand, leading to DNA gaps at expanded repeats. This is in contrast to the single-stranded breaks made by glycosylases that include a single-nucleotide gap – more commonly called a single-strand breaks. We have updated the text to highlight the differences between a nick and a single-strand break.

In addition, the fact that XPA, MSH2, XRCC1, and PARP1 all have an effect on spontaneous CAG repeat contractions but not during nickase-induced contractions suggests that the two processes occur via distinct pathways. We are arguing that this is due to the type of DNA damage that is being repaired at the repeat tract. We have now added this in the discussion: “DNA nicks appear to be repaired by distinct and still poorly understood mechanisms. For example, they stimulate homology-directed repair (HDR) in a human cell-based assay⁵⁹. Intriguingly, this process is suppressed by RAD51 and BRCA2, which are required for homologous recombination at DSBs⁵⁹. PARP inhibition also stimulates nick-induced HDR⁶⁰. Our observation that the same PARP inhibitor has no effect on nickase-induced contraction is suggestive of a different pathway being used at CAG repeats and that DNA nicks are not the mutagenic intermediates leading to nickase-induced contractions. Furthermore, the lack of an effect when knocking down XRCC1 or inhibiting PARP1 implies that the Cas9 nickase leads to contraction via a pathway different from that of BER-generated SSBs. We certainly cannot rule out that DNA nicks lead to contractions independently of the known pathways leading to spontaneous instability. Instead, however, we offer a model (Fig. 5) whereby the Cas9 nickase induces several nicks on the same strand within the repeat tract, thereby generating DNA gaps. This hypothesis is attractive because it provides an explanation for the repeat-length dependency of nickase-induced contractions: shorter repeats have fewer gCTG binding sites and thus DNA gaps are not created as readily, leading to a stable tract. By contrast, it is unclear how nicks or SSBs could explain the length-dependency. Together, these observations suggest that different types of DNA lesions found within the repeat tract are repaired by different pathways, which may dictate the direction of repeat instability.”

Inconsistencies from the literature

Double-strand breaks within the repeat tract caused instability in both directions. The assay does not

clarify or move the field forward. In fact, current reported suggest that DSB cause contraction and or stops instability. The use of Cas9 consistently results in deletion or certainly has a strong deletion bias.

(1) Ye Y, Kirkham-McCarthy L, Lahue RS. The *Saccharomyces cerevisiae* Mre11-Rad50-Xrs2 complex promotes trinucleotide repeat expansions independently of homologous recombination. *DNA Repair (Amst)*. 2016 Jul;43:1-8.

(2) Park CY, Halevy T, Lee DR, Sung JJ, Lee JS, Yanuka O, Benvenisty N, Kim DW. Reversion of FMR1 Methylation and Silencing by Editing the Triplet Repeats in Fragile X iPSC-Derived Neurons. *Cell Rep*. 2015 Oct 13;13(2):234-41).

(3) Sundararajan R, Gellon L, Zunder RM, Freudenreich CH. Sundararajan R, Gellon L, Zunder RM, Freudenreich CH. Double-strand break repair pathways protect against (CAG/CTG repeat expansions, contractions and repeat-mediated chromosomal fragility in *Saccharomyces cerevisiae*. *Genetics*. 201184(1):65-77).

Our intention was not to give the impression that we have uncovered anything new by inducing DSBs and finding that this results in both expansions and contractions. It was, however, a good way to show that our GFP-based assay is consistent with previously published work and thus appears to be a valid readout for CAG repeat instability. We emphasise that our study focusses on how the Cas9 nickase induces predominantly contractions, which is definitely novel. We have used the ZFN to test whether we could obtain expansions from the GFP-based assay. This is because we indeed expected to be able to detect expansions based on the available literature. We used the Cas9 nuclease to test a different way of inducing instability and to be able to compare its effect to that of the Cas9 nickase.

We already had the Park et al and Sundararajan et al papers cited in our manuscript. We have now cited Ye et al in the introduction.

The conclusions that SSB breaks primarily lead to deletions contradict results of many other DNA repair investigators. In mice, loss of XPA and BER enzyme suppress expansions, i.e., indicating that these SSB pathways cause expansion, and there is no evidence that these proteins are involved in DSBs.

(1) Møllersen L, Rowe AD, Illuzzi JL, Hildrestrand GA, Gerhold KJ, Tveterås L, Bjølgerud A, Wilson DM 3rd, Bjørås M, Klungland A. Neil1 is a genetic modifier of somatic and germline CAG trinucleotide repeat instability in R6/1 mice. *Hum Mol Genet*. (2012) 21(22):4939-47 (2012).

(2) Kovtun IV, Liu Y, Bjoras M, Klungland A, Wilson SH, McMurray CT, OGG1 initiates age-dependent CAG trinucleotide expansion in somatic cells. *Nature*, 447(7143): p. 447-52 (2007).

(3) Hubert L Jr, Lin Y, Dion V, Wilson JH. Xpa deficiency reduces CAG trinucleotide repeat instability in neuronal tissues in a mouse model of SCA1. *Hum Mol Genet*. 20(24):4822-3 (2011).

The authors may be correct, but a more in depth analysis beyond the assay needs to be presented to support a divergent point of view or strengthen their conclusions.

We do not conclude that SSBs lead to contractions. Our results argue that SSBs are not involved in nickase-induced contractions. We agree with the reviewer that this is in contrast to BER-induced instability and this is why we propose that DNA gaps are the mutagenic intermediates leading to Cas9-nickase-induced contractions. In addition, the DNA nicks induced by the nickase are very different from the single-nucleotide gaps (i.e., single-strand breaks) that are generated by glycosylases. There is, therefore, no a priori reason to expect the two processes to be similar. That is why we tested the requirements for XRCC1 and PARylation. We have now added a clarification of this along with the Møllersen et al reference that we were missing.

Finally, and important point is that the GFP-based assay agrees with the small-pool PCR experiments (Fig. 2). Together, our results show with confidence that the Cas9 nickase induces more contractions than expansions.

Reviewer #2 (Remarks to the Author):

The authors have adequately addressed the majority of my concerns raised in the original critique of the manuscript. There are just a few minor points / concerns that remain.

1) The PAM sequences for the CAG/CTG repeat targets are not listed in Table S2 as indicated in the authors' response to the reviewers.

The PAM sequences are listed in Table S4. We have now taken this opportunity to add the target sequence and the PAM of the gDM1d gRNA.

2) Given that a suboptimal PAM sequence is being employed for these nucleases/nickases, it is not surprising that the fraction of affected cells (e.g. GFP+) is modest. However for the 12 day treatment shown in Figure 2C where there is a 6% shift in the number of cells in the highest GFP gate but little increase in expansions, I would anticipate that the distribution of the entire population of cells would be shifted toward GFP+. It would be helpful to include the Flow population data for this experiment relative to a 12 day control in the supplement for readers to be able to assess the raw data as in Figure S2A. We have now added representative flow cytometry profiles for 5 and 12 day-treatments in Fig. S2GH. We did not expect the entire population to shift, but rather that there would be a larger tail on the brighter side of the main population. The newly provided profiles demonstrate that nicely.

3) The Figure 2 citations on page 6 need to be adjusted to point to the correct panel. (Fig 2E not 2C)

We have now fixed this and corrected the order of the panels in figure 2 such that they follow the order they are presented in the text.

Reviewer #1 (Remarks to the Author):

The authors have adequately addressed the concerns raised in my revised review.